# Comparison Between Electroporation at Different Voltage Levels and Microinjection to Generate Porcine Embryos with Multiple Xenoantigen Knock-Outs

**DOI:** 10.3390/ijms252211894

**Published:** 2024-11-05

**Authors:** Juan Pablo Fernández, Björn Petersen, Petra Hassel, Andrea Lucas Hahn, Paul Kielau, Johannes Geibel, Wilfried A. Kues

**Affiliations:** 1Institute of Farm Animal Genetics, Friedrich-Loeffler-Institut, 31535 Neustadt, Germany; bjoernpetersen@me.com (B.P.); petra.hassel@fli.de (P.H.); andrea.lucashahn@me.com (A.L.H.); paulkielau@gmx.net (P.K.); johannes.geibel@fli.de (J.G.); 2Graduate School HGNI, University of Veterinary Medicine Hannover (TiHo) Foundation, 30559 Hannover, Germany; 3eGenesis, 2706 County Rd E, Mount Horeb, WI 53572, USA; 4Center for Integrated Breeding Research, University of Göttingen, 37073 Göttingen, Germany

**Keywords:** CRISPR/Cas9, xenotransplantation, microinjection, electroporation, in vitro fertilization, gene editing

## Abstract

In the context of xenotransplantation, the production of genetically modified pigs is essential. For several years, knock-out pigs were generated through somatic cell nuclear transfer employing donor cells with the desired genetic modifications, which resulted in a lengthy and cumbersome procedure. The CRISPR/Cas9 system enables direct targeting of specific genes in zygotes directly through microinjection or electroporation. However, these techniques require improvement to minimize mosaicism and low mutation rates without compromising embryo survival. This study aimed to determine the gene editing potential of these two techniques to deliver multiplexed ribonucleotide proteins (RNPs) to generate triple-knock-out porcine embryos with a multi-transgenic background. We designed RNP complexes targeting the major porcine xenoantigens *GGTA1*, *CMAH*, and *B4GALNT2*. We then compared the development of mosaicism and gene editing efficiencies between electroporation and microinjection. Our results indicated a significant effect of voltage increase on molecule intake in electroporated embryos, without it notably affecting the blastocyst formation rate. Our gene editing analysis revealed differences among delivery approaches and gene loci. Notably, employing electroporation at 35 V yielded the highest frequency of biallelic disruptions. However, mosaicism was the predominant genetic variant in all RNP delivery methods, underscoring the need for further research to optimize multiplex genome editing in porcine zygotes.

## 1. Introduction

Xenotransplantation is a promising solution for the increasing demand for organ transplantation in humans. Due to the close similarity between human and pig organs, domestic pigs have emerged as suitable candidates for transplantable organs [1]. Nevertheless, the presence of xenoantigens limits the success of pig-to-human grafts [2]. To address this challenge, genetically edited pigs carrying xenoantigen knock-outs and expressing human genes to achieve immunotolerance have been generated [3]. Recently, the first clinical xenotransplantations of porcine hearts and a kidney, with several gene knock-outs and multiple immunomodulating transgenes, to human end-stage patients, were reported [4,5,6].

The donor animals were produced via somatic cell nuclear transfer (SCNT), resulting in anti-immunogenic pigs with deletions in the alpha-1,3-galactosyltransferase (*GGTA1*), cytidine monophospho-N-acetylneuraminic acid hydroxylase (*CMAH*), and beta-1,4 N-acetylgalactosaminyltransferase 2 (*B4GALNT2*) genes. These deletions effectively prevent hyperacute rejection and contribute to improved immune compatibility between humans and pigs [7,8]. Nonetheless, SCNT suffers from significant drawbacks. The process requires sophisticated and complex equipment, as well as a high level of expertise, making it labour-intensive, lengthy, and costly. Furthermore, the cloning efficiency of pigs is still low, often failing to exceed an average cloning efficiency of 5%, which tends to diminish as more genetic modifications are introduced into the donor cells [8,9]. Reports also indicate genetic and epigenetic abnormalities in embryos produced by SCNT [10,11], although the consequences for xenotransplantation are unclear.

In recent years, alternative methods to SCNT have gained popularity due to their efficacy and simplicity in gene editing procedures [12]. Microinjection facilitates the delivery of CRISPR/Cas9 plasmids, RNAs, or ribonucleoprotein complexes (RNPs) into oocytes or zygotes. Previously, we reported editing of the pig genome via microinjection of CRISPR/Cas9 plasmids into zygotes [13]. Recently, electroporation has emerged as an alternative to microinjection for the delivery of molecules in embryos [14].

Delivering CRISPR/Cas9 components into embryos, whether via microinjection or electroporation, faces several challenges. These methods often result in low mutation rates and may exhibit a mosaic pattern within the embryos [15]. This can be problematic when wild-type (WT) alleles are present among the cell population. Various studies have explored the optimization of electroporation parameters for pig oocytes and embryos [16,17]. For instance, a study conducted by Navarro-Serna [18] demonstrated that increasing the number of electrical pulses favoured the entry of macromolecules in electroporated oocytes. However, this improvement came at the cost of blastocyst development and did not significantly impact mutation rates.

In our study, we aimed to further improve the efficiency of gene deletion in zygote electroporation by gradually increasing the voltage strength while keeping the pulse poring number and duration constant. Next, we conducted a comparative analysis between the zygote electroporation and microinjection methods. For both techniques, in vitro fertilization (IVF) was carried out utilizing slaughterhouse ovaries and sperm from a multi-transgenic boar [19]. We then compared the gene editing efficiency of these two techniques for the generation of *GGTA1*, *CMAH*, and *B4GALNT2* triple-knock-out pig embryos.

## 2. Results

### 2.1. Voltage Optimization for Enhanced Molecule Delivery in Zygote Electroporation

To assess the optimal electroporation conditions for the initial experiments, we used fluorophore-coupled tetramethylrhodamine–dextran (TMR-D, 70 kDa) for the transfection of parthenogenetically activated embryos. Although the CRISPR/Cas9 ribonucleoprotein complex is approximately 160 kDa, TMR-D was the closest commercially available option to evaluate the electroporation conditions effectively. We established three groups—E30V, E35V, and E40V—representing electroporation at 30 volts, 35 volts, and 40 volts, respectively, to confirm the functionality of the technique and assess blastocyst formation rates (BRs) relative to each voltage. A total of three experiments were performed with duplicates of each experimental group. Successful molecular delivery was confirmed based on the detection of fluorescence twenty-four hours post electroporation (Appendix A). Concerning the developmental potential of zygotes up to the blastocyst stage, no statistical evidence from a *t*-test (assuming approximately normal distribution) could confirm significant differences across the blastocyst rates (BRs). However, it became evident that at 40 volts, the BR was less favourable than that of the other electroporation groups, with its mean decreasing by approximately 13 percentage points compared to the E30V group and 10 percentage points compared to the E35V group (Table 1).

We replicated this experiment in IVF zygotes. Since many reports demonstrate that an increase in voltage levels or pulses results in an elevated level of molecular intake in zygotes [18,20], we aimed to validate this in our experimental approach. Fluorescence was detected in all examined IVF groups and persisted through the blastocyst stage (Figure 1a; Appendix A). ANOVA of our regression model after fluorescence intensity quantification revealed significant effects of experimental day (*p* < 0.01) and voltage (*p* < 0.001) on fluorescence intensity, with an interaction between them being also significant (*p* < 0.05). Overall, this indicates that the fluorescence intensity increased significantly with increasing voltage (Figure 1b), suggesting that the uptake of the reporter is facilitated under these conditions.

BR values in these embryos were comparable with to those measured in parthenogenetically activated embryos (Table 2, Appendix A). We decided to exclude the E40V group for further analyses, as its difference in BR from the other electroporation groups indicated poorer performance.

### 2.2. Guide RNA Selection for Gene Editing

The electroporation protocol initially applied for the delivery of TMR-D in IVF embryos was subsequently employed to target *GGTA1* for knock-out in embryos derived from a heterozygous multi-transgenic boar [19]. Transgene transmission was confirmed by PCR (Appendix A). T7EI analysis conducted on blastocysts resulting from this experiment demonstrated successful editing of the *GGTA1* gene (Figure 2a). Allele-specific sequencing revealed various types of mutations in the GGTA1 1-41 sgRNA targeting region (Appendix A). Subsequently, this sgRNA was combined with sgRNAs targeting *CMAH* (CMAH 10-3 or CMAH 10-4) and *B4GALNT2* (B4GALNT2Ex3-Egen or BGALNT2EX3#1) to create a total of four different RNP combinations for triple gene editing. Both *B4GALNT2* sgRNAs demonstrated success for all four RNP combinations (Figure 2b). However, B4GALNT2Ex3-Egen was chosen for the RNP construct employed in subsequent experiments, as its pairing with CMAH 10-3 along with GGTA1 1-41 suggested greater disruption of the *CMAH* gene than its counterpart (Figure 2c). CMAH 10-4 did not display gene modifications at the *CMAH* locus (Appendix A). Allele-specific sequencing also confirmed the editing of these genes (Appendix A).

### 2.3. Comparative Analysis Between Microinjection and Electroporation Techniques

Following the successful validation of the electroporation experiments, along with confirming the efficiency of the multiplexed RNP complexes in simultaneously knocking out *GGTA1*, *CMAH*, and *B4GALNT2*, three distinct groups were established for a differential analysis between electroporation and microinjection (n = 3). The groups consisted of two electroporation experiment groups, E30V and E35V, as well as a microinjection group. Unfortunately, logistical limitations prevented microinjection from being conducted on the same ovary collection day as electroporation, hindering the statistical comparison of BRs between the two techniques.

#### 2.3.1. Comparisons of Blastocyst Formation Rates Across Electroporation and Microinjection Groups for Gene Editing

Appendix A summarize the BRs resulting from microinjection and electroporation groups, classifying the total number of oocytes involved in each experiment, specifically those subjected to IVF and subsequently selected for further development until the blastocyst stage. The *t*-test analysis revealed a significant difference in the BR between gene-edited microinjection-derived blastocysts and their control (IVF) counterparts (*p* < 0.01). Conversely, no notable differences were observed between electroporation groups in terms of their treatment and voltage level (Table 3).

#### 2.3.2. Comparative Analysis of Mosaicism Development and Mutation Rates in Gene–Edited Blastocysts Among Delivery Technique Groups

Appendix A provides an overview of the electropherograms selected for mutation and mosaicism analysis within each xenoantigen, with a total of n = 173 sequence reads analysed. Appendix A illustrates the genetic variant classification methodology using TIDE, with the xenoantigen *CMAH* as a representative example. Figure 3 illustrates a strong alignment between our genetic variant classification model and the editing efficiencies measured by TIDE for the sequence reads analysed across the three delivery groups.

Chi-squared test-based genetic variant analysis revealed significant differences at both the technique and gene-specific levels (Table 4). Although the variation in unsuccessful editing was pronounced across delivery techniques (*p* < 0.001), with microinjection resulting in the most unsuccessful disruptions, this variation did not extend to comparisons at the gene level. In contrast, the assessment of biallelic disruption showed significant differences between genes (*p* = 0.001), but not between techniques. The *CMAH* guide resulted in the highest number of biallelic disruptions across techniques (n = 16). Mosaicism analysis exhibited significant differences at both the technique (*p* < 0.01) and gene (*p* = 0.01) levels, with the frequency of mosaic *CMAH* edits being the lowest and the frequency of mosaic edits by E30V being significantly greater than those by the microinjection group (*p* < 0.01).

Assuming that the efficiency scores provided by TIDE follow a non-normal distribution, Kruskal–Wallis analysis revealed significant differences based on both technique (*p* < 0.01) and xenoantigen (*p* = 0.01) levels, with no observable interaction effect (*p* = 0.7). However, the Wilcoxon test did not reveal significant differences in specific genes between delivery techniques at the individual gene level (Appendix A).

Finally, we evaluated the number of edited gene loci (mosaic or biallelic) for each blastocyst across the delivery groups (Figure 4). Over fifty percent of blastocysts in both the E30V (65%) and E35V (60%) groups were targeted for gene editing at all three gene loci, while this threshold was not surpassed by that of the microinjection group (45%). Additionally, 25% of the microinjection group displayed no editing of any of the analysed genes, compared to 5% in the E30V group and none in the E35V group. No blastocysts (n = 0) exhibited simultaneous biallelic (homozygous) disruptions across all three gene loci.

## 3. Discussion

This experimental study aimed to develop a protocol for targeting the three major xenoantigens *GGTA1*, *CMAH*, and *B4GALNT2* in IVF-derived porcine embryos originating from the sperm of a multi-transgenic boar through RNP electroporation. The goal was to generate embryos with as many edits as possible, that are suitable for xenotransplantation and surpass the limitations of SCNT. The gene editing efficiency was compared between electroporation and microinjection. Gene editing analysis was conducted at the blastocyst stage, providing insights into the potential gilt genotypes. The results demonstrated that in more than 50% of the analysed blastocysts subjected to electroporation, all three analysed gene loci were targeted (mosaic or biallelic) for gene editing. However, the majority of these edits exhibited a mosaic outcome, demanding further improvements in this procedure.

In examining TMR-D delivery within the E30V, E35V, and E40V groups, the effect of voltage level on molecule intake was noteworthy. However, it is important to consider the significant variations in fluorescence values detected within these groups across experimental days. One possible explanation for this effect between voltage level and molecule intake is that the size of micropores in the oolemma and zona pellucida increases with an increase in the applied voltage, facilitating the influx of molecules into the embryo. An increase in fluorescence intensity is also observed when the number of electrical pulses increases under constant voltage levels [18]. Nevertheless, it is noteworthy that these experiments were performed with TMR-D 3000MW, which is considerably smaller than the size used in our research. Similar results are applicable in bovine and rat experiments [20,21].

The low blastocyst development rate observed in the E40V group was not significantly different from that in the E30V, E35V, or IVF control groups. However, there was a decrease of approximately 10% in both the parthenogenetic and IVF experiments compared to these groups. In a separate study, it was demonstrated that for porcine zygotes subjected to electroporation at 40 V, both the blastocyst rate and cleavage rate were significantly lower (*p* < 0.05) than those of zygotes electroporated at less than 30 V/mm, demonstrating low blastocyst viability at this voltage level [17]. We were unable to perform microinjection and electroporation on the same experimental days. Consequently, due to the significant observed effects of the experimental day demonstrated previously, it was not possible to conduct a statistical analysis of blastocyst formation rates between these two techniques. However, it became evident that microinjection had a detrimental effect on BR, at least when compared to the non-microinjected controls. This trend was not observed in the E30V and E35V groups, which, similar to TMR-D delivery experiments, exhibited only minor differences among themselves or with their controls. Further literature studies comparing these two techniques for determining blastocyst formation rates in pigs revealed divergent outcomes. In a study conducted by Navarro-Serna et al. [18], no significant difference was observed between the two techniques. However, in a study conducted by Le et al. (2021) [22], microinjection experiments in porcine embryos resulted in a notably lower blastocyst development rate than electroporation. These discrepancies highlight how microinjection performance in terms of blastocyst formation rates can significantly vary from person to person. In other animal studies, such as those involving rats and mice, the survival rate of embryos was greater after electroporation than in microinjection experiments. In mice, not only did blastocysts exhibit enhanced viability and development, but there was also a significant increase in the percentage of pups born per total embryos harvested [23,24]

Concerning the assembly of our multiplexed RNP complex, T7EI analysis did not reveal any gene modifications at the *CMAH* locus when CMAH 10-4 sgRNA was delivered in combination with other sgRNAs within an RNP complex. Additionally, compared with B4GALNT2Ex3-Egen, BGALNT2EX3#1 seemed to affect the performance of CMAH 10-3. These outcomes could be attributed to variations in the efficiencies of sgRNAs for gene editing or potential cross-linking between them, specifically when a sgRNA affects the editing efficiency of others within an RNP complex. This becomes a concern when attempting to generate multiple edited embryos in one step. For instance, a study by Hirata et al. (2020) [25] demonstrated that pooled gRNAs targeting multiple genes reached lower mutation efficiencies than those achieved after electroporation with single gRNAs. Another study conducted by Sakurai et al. (2020) [26] compared the multiplexing capability of two different electroporation methods with the aim of disrupting up to ten genes. The study demonstrated that while one method exhibited higher genome editing efficiency than the other, this observation was applied only to specific sgRNAs that were analysed. This finding aligns with our results, which suggest varying outputs in editing efficiency depending on the targeted gene locus (Appendix A). Overall, it is critical to conduct an appropriate selection of sgRNAs within multiplexed RNPs, in order to successfully edit multiple genes in one step.

When comparing our three different delivery groups, considering all analysed sequences regardless of the specific xenoantigen that was sequenced (Appendix A), no significant difference in means of indel score values was observed between electroporation at 35 and 30 volts (Wilcoxon test). Nonetheless, the indel scores associated with microinjection were significantly lower than those observed in both electroporation experiments. Animal studies indicate that mutation efficiencies produced through microinjection are either not significantly different or, in a few cases, worse than those achieved by electroporation [18,23,24,27]. Microinjection also resulted in more unsuccessful edits compared to both electroporation procedures, a pattern not observed when unsuccessful edits across genes were examined. This difference may be due to the potential failure of microinjection to deliver RNPs to specific embryos. This becomes more evident when comparing the number of edited alleles per blastocyst. In the microinjection group, five blastocysts showed no mutations in any of their targeted genes, whereas this was observed in only one blastocyst of the E30V group and none in the E35V group. This observation underscores the advantage of electroporation over microinjection, where not only can embryos be edited in bulk but also the vast majority of them can be effectively targeted with RNP delivery. Similar findings were reported in multiplexing studies in pigs conducted by Takeshige Otoi’s group, where electroporation alone successfully targeted all intended genes in most blastocysts [28,29]. In contrast to our group, Takeshige Otoi’s group achieved homozygous knock-out embryos for all xenoantigens. While there are slight differences in protocol—such as the number of embryos per run, reagent concentration, and electroporation device and settings—the key distinction is the sgRNA screening process. This is further emphasized by the fact that our microinjection experiments also failed to produce homozygous mutations. Our group limited the search to a maximum of two sgRNAs per gene locus, whereas Otoi’s group expanded this search and employed TIDE for assessment. Compared to the conventional T7 endonuclease assay, TIDE allows for more accurate detection of homozygous mutations, enabling better sgRNA selection and thus reducing mosaicism in direct zygote gene editing.

Although an increase in voltage significantly improved TMR-D intake in zygotes, no substantial differences in gene editing or genetic variant outcomes were observed between E30V and E35V. Similarly, increasing the frequency of electrical pulses did not increase mutation rates in another study [18]. This trend might be related to the size of the Cas9 protein (nearly 160 kDa), which becomes inaccessible to zygotes regardless of the voltage level or pulse frequency. However, it is worth noting that, although not statistically significant, the amount of biallelic disruptions observed in the E35V group was always greater than in the E30V group without compromising blastocyst rate performance. In mouse studies, the use of maternally expressed Cas9 bypassed the limitations of Cas9 supply, which resulted in increased levels of mutagenesis and homology-directed repairs [26].

Mosaicism persists as a limitation observed in cloning-free procedures among pigs and other animals [28,30]. This limitation is critical due to the genetic variability in cell populations, which negatively impacts the germ-line transmission of mutations to offspring. Additionally, multiplexing adds a further complication to this topic, making the production of triple-knock-out pigs a challenging endeavour. Our results demonstrated that mosaicism was the most recurrent genetic variant across sequences regardless of the gene locus or delivery technique. In this context, electroporation would be particularly suited for single gene editing, as this approach minimizes the likelihood of mosaicism when fewer genes are targeted, thereby increasing the chances of obtaining non-mosaic specimens. However, for multiplexing studies, further breeding of mosaic animals may ultimately produce the desired homozygous genotype. Studies in mice and buffalo aiming to mitigate mosaicism suggest the introduction of RNPs at early embryo stages [31,32]. In pigs, even microinjection of CRISPR/Cas9 before insemination has proven to be effective in preventing mosaicism [33]. Conversely, electroporation or microinjection at the two-cell stage increases the risk of mosaicism [27]. Further studies also demonstrated that an increase in Cas9 concentration led to a decrease in mosaic outcomes, which came at the cost of low blastocyst rates [28]. However, neither of these strategies could completely alleviate mosaicism, or in some cases, only a single gene was studied for its disruption. With that being said, a balance needs to be found among the various factors that positively influence gene editing. Moreover, additional assessment is required for other factors affecting gene editing that may not necessarily be related to laboratory procedures but could still impact the overall outcome, potentially being oocyte-related.

In conclusion, this study compared cloning-free methods for generating multiplexed gene-edited embryos using transgenic boar sperm. Electroporation at 35 volts proved to be more advantageous than at 30 V by facilitating effective molecule entry and slightly improving rates of biallelic disruptions, without compromising embryo viability. Conversely, microinjection was found to negatively impact embryo viability and resulted in higher rates of unedited genes compared to both electroporation techniques. Overall, electroporation emerged as a method for efficient and effective gene editing in embryos.

## 4. Materials and Methods

### 4.1. Study Design

#### 4.1.1. Optimizing Voltage Levels for Efficient Molecule Delivery into Zygotes via Electroporation

The initial goal of our study was to identify suitable embryo electroporation parameters employing the Nepa 21 electroporator (Nepa Gene, Chiba, Japan). Accordingly, we introduced the red fluorescent compound tetramethylrhodamine 70 kDA–dextran (TMR-D) to groups of 30 parthenogenetically activated and IVF zygotes at 30, 35, and 40 volts. These groups were denoted as E30V, E35V, and E40V. Subsequently, we assessed their fluorescence intensity, in vitro development, and blastocyst formation rates (BRs). This process aimed to establish a protocol for the voltage-specific electroporation of RNP complexes into IVF embryos.

#### 4.1.2. Assembly of a Multiplexed RNP Complexes Targeting *GGTA1*, *CMAH* and *B4GALNT2* in Electroporation and Microinjection Experiments

We explored different combinations of single guide (sg)RNAs to assemble a multiplexed RNP complex capable of simultaneously targeting the following porcine xenoantigens: *GGTA1*, *CMAH*, and *B4GALNT2*. The resulting RNP complexes were then subjected to both electroporation and microinjection for comparative gene editing analysis. To assess disruption efficacy, pooled gene-edited blastocysts (corresponding to a specific RNP complex mixture) were subjected to T7 endonuclease assays (T7EI). Subsequently, we conducted allele-specific sequencing to confirm disruption at the RNP target site for each xenoantigen.

#### 4.1.3. Comparative Analysis Between Microinjection and Electroporation Techniques in Terms of Blastocyst Developmental Potential, Editing Efficiencies, and Mosaicism Development

The electroporation groups and a group subjected to microinjection were selected for the delivery of our RNP complex into IVF-derived embryos. In addition to analysing blastocyst formation rates, single blastocysts from each group were subsequently analysed for genetic variant outcomes and gene editing efficiencies based on allele decomposition analysis using TIDE software (version 3.0.0) [34].

### 4.2. Ovary Collection and Maturation

Porcine ovaries were obtained from a local slaughterhouse and underwent initial cleansing in a 0.9% NaCl (Roth #3957.2, Karlsruhe, Germany) solution supplemented with 0.06 g/l penicillin (AppliChem #A1837, Darmstadt, Germany) and 0.131 g/L streptomycin (AppliChem #A1852,0250) before oocyte retrieval. Aspiration of the oocytes was carried out using a house vacuum connected to a Falcon tube. Subsequently, the collected oocytes were washed in porcine X Medium (PXM) [35]. Adequate cumulus–oocyte complexes (COCs) were defined by uniform cytoplasm and several layers of cumulus cells encompassing the zona pellucida. For maturation, the oocytes were transferred into FLI maturation medium [35]. Each collection involved the incubation of a minimum of 300 COCs for 40–44 h at 38 °C in a humidified air environment with 5% CO_2_.

### 4.3. Parthenogenetic Activation

After maturation, the COCs were exposed to a solution of TL (Tyrode’s lactate)–HEPES (Roth #9105.0) + 0.1% hyaluronidase solution (Sigma #H3506-1G, San Luis, MO, USA) for one to two minutes to remove cumulus cells. Denuded oocytes were washed in TL-HEPES 296 Calcium Ca [35], and only those with visible polar bodies (metaphase II oocytes) were selected for activation. Oocytes were transferred to an equal mixture of TL-HEPES 296 Calcium and activation medium (Ca2+ Sor2 medium) [35]. For electrical activation, oocytes were transferred into an activation medium, and a 24 V pulse for 45 μs in a 0.2 mm Multiporator chamber (Eppendorf Multiporator, Hamburg, Germany) was applied. Additionally, chemical activation was performed by pipetting oocytes into wells containing PZM (PZM-5; Research Institute for the Functional Peptides Co., Yamagata, Japan) + 2 mM 6-DMAP (Sigma #D2629) and incubating them at 38 °C. Subsequently, the cells were transferred to PZM and placed in an incubator at 39 °C.

### 4.4. Extraction, Freezing, and Thawing of Porcine Semen

Sperm was obtained from a multi-transgenic boar heterozygously carrying clustered human transgenes known to suppress complement system activation (*hCD46*, *hCD55*, *hCD59*, *hA20*, and *hHO-1*) within the Rosa26 locus [19] using the hand-gloved method with a phantom, filtered through gauze, and diluted 1:1 with prewarmed (38 °C) Androhep^®^plus extender (Minitube, Tiefenbach, Germany). The diluted semen was transferred to the laboratory and kept in 50 mL centrifugation tubes at room temperature for 60 min. The semen was then cooled to 15 °C over 90 min in an incubator. Next, the tubes were centrifuged at 800× *g* and 15 °C for 10 min to reduce seminal plasma. The supernatant was discarded and replaced with a chilling extender (80% 322 mM lactose solution + 20% egg yolk) equilibrated at 15 °C. The sperm concentration was adjusted to 1.8 billion spermatozoa/mL using a NucleoCounter (ChemoMetec A/S, Allerod, Denmark). The semen was cooled to 5 °C over 90 min and maintained at 4 °C for 30 min.

All further steps were conducted at 4 °C. Chilled semen was mixed 2:1 with a freezing extender (92.5 mL chilling extender, 1.5 g Orvus ES paste (OEP, Minitube), 6.0 g glycerol) to achieve a final concentration of 1.2 billion sperm/mL and 1.74% glycerol. Straws (0.25 mL; Minitube) were filled and sealed with an MPP Uno machine (Minitube). Freezing was performed in a Styrofoam box with liquid nitrogen. The samples were placed on metal racks 4 cm above the liquid nitrogen for 20 min, then plunged into liquid nitrogen and stored in cryo containers. Straws with frozen semen were plunged in a water bath at 37 °C for 17 s. Then, the outer surface of the straws was carefully dried, the straws were cut open, and the content was expelled into a 3 mL Androhep-filled 15 mL Falcon tube. Centrifugation took place at 867× *g* for three minutes at 30 °C, and the pellet was washed again with 3 mL Androhep before being resuspended in 0.5 mL Fert-Talp [36].

### 4.5. In Vitro Fertilization (IVF)

The matured oocytes were coincubated for 5 h with frozen–thawed ejaculated spermatozoa (1 × 10^6^ cells/mL) in porcine fertilization medium [36], at a ratio of 60 spermatozoa to one oocyte. Then, the embryos were cultured in PZM in a humidified incubator at 39 °C with 5% CO_2_ until the cytoplasmic maturation, pronuclear formation, and the introduction of the Cas9–guide RNA (gRNA) ribonucleoprotein complex (RNPs) through electroporation or cytoplasmic microinjection took place. This occurred 15 h post-IVF or parthenogenetic activation.

### 4.6. Electroporation

For TMR-D electroporation, tetramethylrhodamine–dextran (70,000 MW, Thermofisher, Waltham, MA, USA) was diluted in TL-HEPES to a concentration of 2 mg/mL. For RNP electroporation (all components purchased from IDT, Coralville, IA, USA), we assembled the multiplexed RNP constructs in accordance with the protocol of Tröder et al. (2018) [24], by diluting 4 µM of the respective sgRNAs (Appendix A) and 4 µM Cas9 along with 10 µM Alt-R CRISPR/Cas9 electroporation enhancer, in OptiMem (Thermofisher #31985062).

Groups of 30 embryos each were washed three times in OptiMem before being placed in a chamber (CUY505P5, Nepa Gene; Chiba, Japan) containing 20 μL of the TMR-D or RNP solution. Electroporation was performed in a Nepa 21 electroporator (Nepa Gene); the parameters were set as outlined in Appendix A and poring voltages were set at 30, 35, and 40 volts. After electroporation, the zygotes were removed from the chamber using a glass capillary pipette, followed by a triple wash in TL-HEPES and an additional triple wash in PZM for subsequent culture.

### 4.7. Cytoplasmatic Microinjection

Multiplexed RNP constructs for microinjection were generated by diluting sgRNA and Cas9 components in TE buffer consisting of 10 mmol/L Tris–HCl (pH 7.6) and 0.25 mmol/L EDTA pH 8.0 to a final individual concentration of 2 µM, following a previously published CRISPR editing protocol [37].

Individual zygotes were fixed by suction to a holding pipette, while the injection capillary was inserted into the cytoplasm through the zona pellucida and the cell membrane. Approximately 10 pL of the multiplexed RNP solution was injected with a pressure of 400–600 hPA into the zygote cytoplasm using a pressure-controlled Eppendorf Transjector 5246 (Eppendorf, Hamburg, Germany).

### 4.8. Evaluation of Fluorescence Intensity in Embryos Post TMR–Dextran Delivery

The fluorescence intensity emitted by pig zygotes, resulting from the delivery of TMR-D via electroporation, served as a reference for molecular intake in pig embryos. This was documented twenty-four hours post electroporation using an LED Inverted Fluorescence Phase Contrast Microscope (Leica, Wetzlar, Germany). A built-in Texas Red (TXR) fluorescence filter (Ex: 560/40 nm, DM: 595 nm, EM: 630/75 nm) was employed to detect red-fluorescent embryos under a 10× magnification and 3 s exposure time. Images were subsequently analysed by ImageJ (version 1.48) [38] to quantify fluorescence values. This quantification involved subtracting the fluorescence intensity emitted from zygotes from the fluorescence of the background.

The linear regression model utilized to evaluate the effects of voltage and experimental days on the fluorescence intensity resulting from the zygotes is indicated as follow:$$y_{ij}=\mu +Day_{i}+\beta_{1}*Voltage+Day_{i}*\beta_{2}*Voltage+e_{ij}$$

In this equation, y represents the fluorescence intensity, *μ* the overall intercept, Day is the *i*-th experimental day (*i* = 1, 2, 3), the voltage is the applied voltage, *β*_1_ and *β*_2_ are regression coefficients, and *e*_*i**j*_ represents the residual error. The effects in the model were tested for significance using an Analysis of Variance (ANOVA) on the linear model in R version 4.3.0 [39].

### 4.9. Analysis of Target Sites

#### 4.9.1. DNA Extraction of the Blastocysts

On day 6 post-IVF, the number of blastocysts per group was recorded. Subsequently, blastocysts were individually transferred into PCR nuclease-free tubes, containing 15 μL of cell lysis buffer containing 0.02% SDS, 20 mM Tris–HCl (pH 8.4), and 0.05 mg/mL proteinase K. The lysis process took place at 56 °C for one hour followed by 10 min at 95 °C. Lysed blastocysts were stored at −20 °C for further analysis.

#### 4.9.2. PCR Analysis of Targeted Genes

The *GGTA1*, *CMAH*, and *B4GALNT2* genes were individually amplified in pooled blastocysts (study 2) or single blastocysts (study 3). Additionally, the transgene hA20 corresponding to the multi-transgenic cassette carried in our boar intended for IVF [19] was detected in single blastocysts. For each primer pair (Appendix A), 5 μL of the corresponding lysate was subjected to PCR under the following conditions: 94 °C for 2 min, followed by 40 cycles of 94 °C for 30 s, annealing at primer-specific temperature for 45 s, 72 °C for 30 s, and a final extension at 72 °C for 5 min. To evaluate the success of the experiment and validate positive and negative (water only) controls, 5 μL of the PCR mixture was run on a 1% agarose gel. PCRs with faint bands or those that could not be properly sequenced were subjected to reamplification through a 12-cycle PCR run. Signals were visualized by UV exposure with the Fusion SL4-3500.WL (Vilber) and the corresponding software FusionCapt V15.18 using an exposure time of 2.5 s. ImageJ (version 1.48) was employed to observe the images at their original exposure time and for processing.

#### 4.9.3. T7 Endonuclease Assay

A total of 12 µL amplified, non-purified PCR products were subjected to T7 endonuclease assay (NEB, Ipswich, MA, USA) following the manufacturer’s protocol. Subsequently, the reaction was quenched with 1.5 μL of 0.25 M EDTA. The fragmented product was then analysed via gel electrophoresis at 80 V, for 60 min on a 1.5% agarose gel, followed by UV light exposure.

#### 4.9.4. Allele-Specific Sequencing of *GGTA1*, *CMAH*, and *B4GALNT2*

For haplotype identification, PCR products were subcloned and inserted into the pGEM-T Easy Vector system (Promega, Walldorf, Germany) following the manufacturer’s protocol. Subsequently, the cloned DNA was transformed into XL10 bacteria. The transformed bacteria were cultured on agar dishes supplemented with ampicillin (100 μg/mL) for 16 h. To ensure robust sequencing, twelve colonies were selected and subjected to sequencing using the T7prom primer.

#### 4.9.5. Sanger Sequencing

A total of 15 μL from each PCR reaction was transferred to a 96-cell non-skirted plate. Subsequent PCR clean-up and direct sequencing procedures were conducted by LGC Genomics GmbH (Berlin, Germany). Failed electropherograms or those displaying poor base call quality across the entire sequence were excluded from the analysis.

#### 4.9.6. Analysis of Genomic Edits

A total of 20 blastocysts resulting from each delivery group were collected and each of the targeting RNP regions was amplified for further Sanger sequencing. The resulting electropherograms were then subjected to TIDE analysis. TIDE is a software designed to computationally breakdown sequence traces [34]. This tool identifies and quantifies the frequencies of indel sizes within a population, determining the effectiveness of templated genome editing. Genetic variant classification methodology using TIDE was analysed as follows: biallelic disruption was identified when the targeted mutation was detected, and the presence of the wild-type (WT) allele was absent in the analysed sequence. Conversely, unsuccessful editing manifests as the WT genotype, showing no traces of the targeted mutation. Mosaicism, characterized by a mix of mutant and wild-type alleles within the spectrum of indels, can be challenging to distinguish from heterozygosity based on Sanger sequencing alone. Consequently, we use the term “mosaic” to refer to both scenarios. A chi-squared test was conducted to identify differences in genetic variants across techniques and gene targeting guides using R [39].

## Figures and Tables

**Figure 1 ijms-25-11894-f001:**
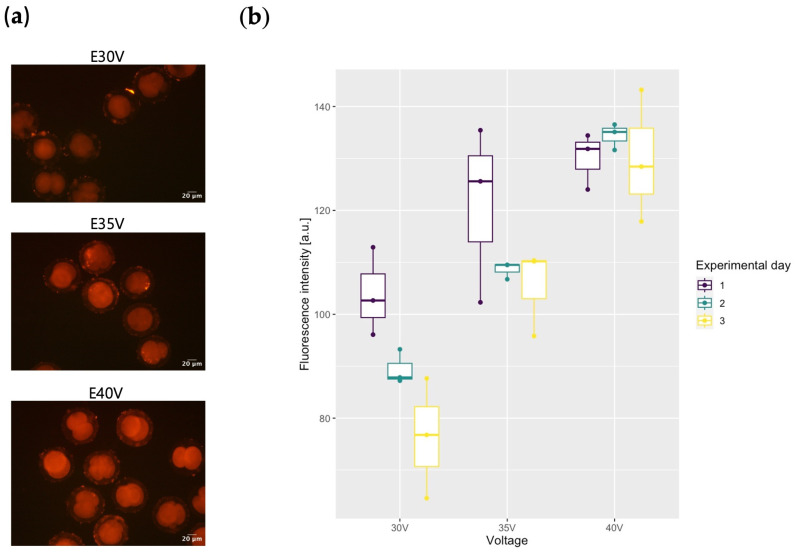
Electroporation-mediated TMR–Dextran delivery into porcine zygotes. (**a**) Normalized images of TMR-D delivered to embryos electroporated with different voltages (10× magnification). (**b**) Quantification of fluorescence values 24 h post electroporation reveals significant effects among experimental groups. Day: *p* < 0.01, voltage: *p* < 0.001, day × voltage: *p* < 0.05. Original recordings are displayed in Appendix A.

**Figure 2 ijms-25-11894-f002:**
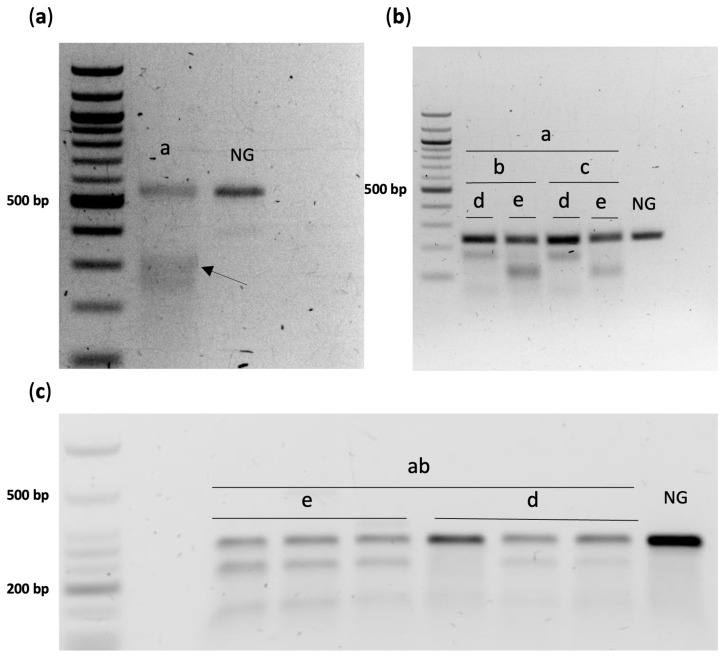
Visualization of T7 endonuclease I (T7EI) activity on PCR products from pooled blastocysts. (**a**) The presence of T7E1-digested fragments (arrow) confirms CRISPR/Cas9-induced mutations in blastocysts resulting from *GGTA1* RNP delivery via electroporation. (**b**) Editing of *B4GALNT2* for diverse RNP combinations (abd, abe, acd, ace). (**c**) The digestion of *CMAH*, conducted in triplicate experiments, reveals a stronger digestion band in the presence of B4GALNT2Ex3-Egen (abe) within the RNP compared to BGALNT2EX3#1 (abd). Original figures are displayed in Appendix A. The sgRNAs utilized in RNP delivery are as follows (a–e): a. GGTA1 1-41; b. CMAH 10-3; c. CMAH 10-4; d. BGALNT2EX3#1; e. B4GALNT2Ex3-Egen; NG = no sgRNA delivered.

**Figure 3 ijms-25-11894-f003:**
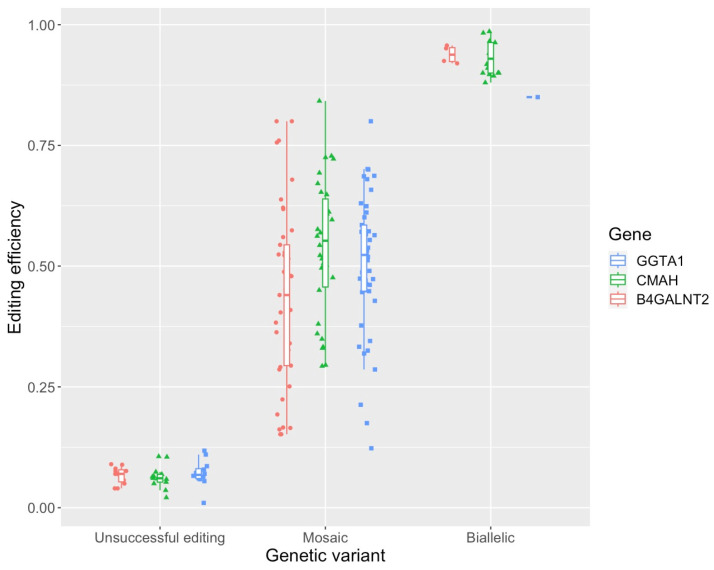
Alignment between genetic variant and editing efficiency in analysed blastocysts. Each data point represents a sequence electropherogram for either *GGTA1*, *CMAH*, or *B4GALNT2* genes. These data are derived from blastocysts produced through either E30V, E35V, or microinjection groups. The data were filtered through TIDE software (version 3.3.0) for precise classification of editing efficiency and genetic variant.

**Figure 4 ijms-25-11894-f004:**
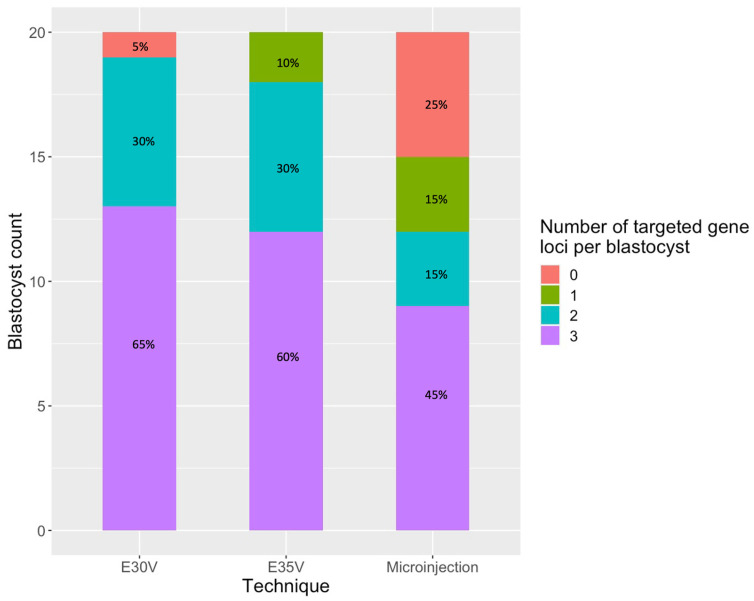
Proportion of edited gene loci in single blastocysts across RNP delivery techniques. Each column’s colours represent the number of RNP targets edited (mosaic or biallelic) in each blastocyst across the different RNP delivery techniques. Blastocysts where no xenoantigen was edited within a specific RNP delivery technique are represented in red.

**Table 1 ijms-25-11894-t001:** Analysis of BR for E30V, E35V, and E40V groups for TMR–Dextran delivery in parthenogenetically activated embryos. No statistical evidence of differences in the mean BR were found using a *t*-test.

Condition	N° of Replicates per Trial	Zygotes	Blastocysts	Mean BR (SD)
E30V	2	172	61	34.88 ^a^ (4.8)
E35V	2	164	52	31.46 ^a^ (7.8)
E40V	2	170	37	21.5 ^a^ (6.6)
Control (parthenogenetic)	1	83	29	34.67 ^a^ (8.8)

Note: Three independent trials were carried out, ^a^: no statical difference across groups.

**Table 2 ijms-25-11894-t002:** Analysis of BR at E30V, E35V, and E40V for TMR–Dextran delivery in IVF embryos. No statistical evidence of differences in the mean BR was found using a *t*-test.

Condition	Zygotes	Blastocysts	Mean BR (SD)
E30V	92	33	36 ^a^ (8.2)
E35V	89	30	34 ^a^ (6.7)
E40V	88	22	25 ^a^ (7.3)
Control (IVF)	134	38	29 ^a^ (9.6)

Note: three independent trials were carried out, ^a^: no statistical difference across groups.

**Table 3 ijms-25-11894-t003:** Analysis of BR in the E30V, E35V, and microinjection groups for RNP delivery.

Technique	Treatment	Voltage	Total Zygotes ^1^	Total Blastocysts	Mean BR (SD)
Electroporation	IVF Control	-	142	58	40 (14)
	Gene-edited	30 V	235	65	28 (5)
	Gene-edited	35 V	238	58	25 (8)
Microinjection	IVF Control	-	142	76	52 ^a^ (11)
	Gene-edited	-	207	46	22 ^b^ (14)

^1^ Zygotes collected from three independently performed experiments; superscripts (letters) denote significant differences (*p* < 0.01) as determined by *t*-test.

**Table 4 ijms-25-11894-t004:** Genetic variant classification for *GGTA1*, *CMAH*, and *B4GALNT2* across RNP delivery techniques.

Genetic Variant (GV)	Delivery Group (DG)	B4GALNT2	CMAH	GGTA1	Total of GV by DG
UE	E30V	3	2	2	7 ^b^ (19%)
	E35V	3	2	2	7 ^b^ (19%)
	MI	5	9	8	22 ^a^ (61%)
	Total	11	13	12	36
Mosaic	E30V	15	14	17	46 ^a^ (40%)
	E35V	14	9	16	39 (34%)
	MI	12	7	12	31 ^b^ (27%)
	Total	41 ^1^	30 ^2^	45 ^1^	116
Biallelic Disruption	E30V	1	4	0	5 (24%)
	E35V	2	8	1	11(52%)
	MI	1	4	0	5 (24%)
	Total	4 ^2^	16 ^1^	1 ^2^	21

UE: unsuccessful editing, MI: microinjection. Superscripts indicate significant differences across techniques (letters) and genes (numbers) as determined by chi-squared test.

## Data Availability

The data generated and/or analysed during the current study are available from the corresponding author upon reasonable request. Sequencing data were deposited at Zenodo under CRISPRCas9 B4GAL CMAH GGTA1.zip and Subcloning Assays B4G CMAH GGTA1.zip: https://zenodo.org/records/13909317?token=eyJhbGciOiJIUzUxMiJ9.eyJpZCI6ImU1MWViYjYxLTRmZTYtNDUwZi05MmZlLWE5YjRlZDE3NzA3MiIsImRhdGEiOnt9LCJyYW5kb20iOiI4NDEyNjE5YTFjNTY2M2NlYjY5ZDNjODBlYWZlMGM4YiJ9.24A4Jw4TTHPeuxSHnweWUtH6b0MjfPjB4VD1iDaKKH7_cF2FB0FBLvI7h-9d1R7ZEDdOtN3IaoNRqIufwpREDA, accessed on 9 October 2024.

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
