# Peer review of "Comparison Between Electroporation at Different Voltage Levels and Microinjection to Generate Porcine Embryos with Multiple Xenoantigen Knock-Outs"

_ijms, 2024, doi:10.3390/ijms252211894_

Round 1

Reviewer 1 Report

Comments and Suggestions for Authors

The paper presents a study to further improve a novel technique to generate multiple knock-out pigs via CRISPR/Cas9. The study design is well structured. The continuous text still contains correction notes in several positions "(Error! Reference source not found)", which should be removed . It's not clear if they have already been adressed or not. The values on the percentage decline in blastocyst rate of the 40 volts group compared to 30 volts and 35 volts (p2 line 92) are not correct, at least not supported by the values given in Table 1.
Some typos: p3 l 104 and 109 "porcinea", porcineb"; p 5 l 164 no indication which figure is meant. Not all figures in supplementary material are referred to in the text; brief explanations of all figures would simplify understanding.

I would recommend to clarify for which objective the electroporation technique should be used. Should the gene-edited pigs produced this way directly be used as organ donors in the future, or be raised as founder animals for a genetically modified breeding herd? This statement is crucial to evaluate if the high percentage of mosaicism/monoallelic knockouts is somehow acceptable or not.

Author Response

Response to Reviewer 1: Thank you for the valuable feedback and suggestions to improve our manuscript. We also apologize for the errors that occurred during the upload and formatting of the Word document; all issues have now been addressed.

Comment 1: The paper presents a study to further improve a novel technique to generate multiple knock-out pigs via CRISPR/Cas9. The study design is well structured. The continuous text still contains correction notes in several positions "(Error! Reference source not found)", which should be removed. It's not clear if they have already been addressed or not.

Response 1: We apologize for these overlooked errors. We have conducted a thorough review of the manuscript to ensure that all “(Error! Reference source not found)” have been removed, and all references are now correctly displayed.

Comment 2: The values on the percentage decline in blastocyst rate of the 40 volts group compared to 30 volts and 35 volts (p2 line 92) are not correct, at least not supported by the values given in Table 1.

Response 2: Thank you for pointing this out. We have revised the terminology from "percentage" to "percentage points" as highlighted in the manuscript 

Comment 3: Some typos: p3 l 104 and 109 "porcinea", "porcineb"; p5 l 164 no indication which figure is meant.

Response 3: We have adjusted the document formatting to align fully with the IJMS template. Figures are now correctly specified as highlighted in the manuscript.

Comment 4: Not all figures in the supplementary material are referred to in the text; brief explanations of all figures would simplify understanding.

Response 4: We have revised the text to ensure that all figures in the supplementary material are referenced within the main manuscript. Brief explanations for each figure have also been added to enhance comprehension.

Comment 5: I would recommend clarifying the objective for which the electroporation technique should be used. Should the gene-edited pigs produced this way directly be used as organ donors in the future, or be raised as founder animals for a genetically modified breeding herd? This statement is crucial to evaluate if the high percentage of mosaicism/monoallelic knockouts is somehow acceptable or not.

Response 5: Thank you for this suggestion. We have added a statement in the Discussion section to clarify the objective of using the electroporation technique and to explain the intended use of the gene-edited pigs produced

p343-347: "Our results demonstrated that mosaicism was the most recurrent genetic variant across sequences regardless of the gene locus or delivery technique. In this context, electroporation would be particularly suited for single gene editing, as this approach minimizes the likelihood of mosaicism when fewer genes are targeted, thereby increasing the chances of obtaining non-mosaic specimens. However, for multiplexing studies, further breeding of mosaic animals may ultimately produce the desired homozygous genotype"

This clarifies that electroporation is optimal for single-gene edits, as the probability of mosaicism reduces. However, mosaic animals from multiplexing studies could be used as founders in breeding programs to achieve the homozygous genotype.

Reviewer 2 Report

Comments and Suggestions for Authors In this study, the authors designed ribonucleotide proteins (RNPs) complexes targeting the major porcine xenogens GGTA1, CMAH, and B4GALNT2. Chimeric development and gene editing efficiency between electroporation and microinjection were then compared. Results compared cloning-free methods using transgenic boar sperm to generate multiplex genetically edited embryos. Electroporation at 35 volts is more advantageous than 30 volts because it facilitates efficient entry of molecules and slightly increases the rate of biallelic destruction without affecting embryonic viability. Conversely, microinjection was found to negatively affect embryo viability and result in a higher rate of unedited genes compared to the two electroporation techniques. Comments: 1. In Figure 1, scale bar is needed for panel (a). 2. There are a lot of “(Error! No reference source found)” throughout the manuscript. Please have a check and make them correct. 3. Make Table 4 well organized like other tables. 4. Did the authors make statistics for the different groups/trials in four tables? Give signs please.

Author Response

Thank you for the valuable feedback and suggestions provided to improve our manuscript. Below, we address each comment in detail and describe the corresponding changes made. We also apologise for the errors occurred at the moment of uploading and formatting the word document.

Comment 1: In Figure 1, scale bar is needed for panel (a).

Response 1: Thank you for noting this. We have added a scale bar to panel (a) in Figure 1 for all three recordings as highlighted in the manuscript.

Comment 2: There are a lot of “(Error! No reference source found)” throughout the manuscript. Please have a check and make them correct.

Response 2: We apologize for the oversight. All “(Error! No reference source found)” issues throughout the manuscript have been corrected. References now display accurately. 

Comment 3: Make Table 4 well-organized like other tables.

Response 3: Thank you for this suggestion. When transferring Table 4 into the IJMS template, some superscripts were not properly formatted, affecting the table's clarity. We have revised and reformatted Table 4 to ensure it is consistent with the other tables in the manuscript.

Comment 4: Did the authors make statistics for the different groups/trials in the four tables? Give signs please.

Response 4:  Statistical analyses were indeed conducted for the different groups/trials across the tables. We have now included appropriate statistical significance markers in the form of superscripts in all four tables to clarify these results.